Utilizing loop-mediated isothermal amplification (LAMP) for detecting hemoglobin Constant Spring and hemoglobin Pakse mutations amidst the high prevalence and genetic heterogeneity of thalassemia in Thailand

Wongprachum Kasama 1
Thitipoomdecha Nichakan 2
Ananratanakit Phakkamon 2
Prakobkul Wattanakit 2
Angkuranak Unchasa 2
Sawangkul Nitchagan 2
Panichchob Prapaporn 3
Karnpean Rossarin 3 4 5
Jomoui Wittaya wittayaj@g.swu.ac.th 3 4 5
1 Nutritional Science, Dietetics and Food Safety Program, Faculty of Public Health, Mahasarakham University , Maha Sarakham , Thailand
2 Faculty of Medicine, Srinakharinwirot University , Ongkharak , Nakhon Nayok , Thailand
3 Department of Pathology, Maha Chakri Sirindhorn Medical Center, Faculty of Medicine, Srinakharinwirot University , Ongkharak , Nakhon Nayok , Thailand
4 Clinical Research Center, Faculty of Medicine, Srinakharinwirot University , Ongkharak , Nakhon Nayok , Thailand
5 Research Cluster in Hematology and Genetic Diseases, Faculty of Medicine, Srinakharinwirot University , Ongkharak , Nakhon Nayok , Thailand
Beddoe Travis
Electronic publication date: 2025 Jul 7
Publication date: 2025
Volume: 13
Electronic Location ID: e19687
Received 2025 Feb 25; Accepted 2025 Jun 11
Copyright: ©2025 Wongprachum et al.
Copyright year: 2025
Copyright holder: Wongprachum et al.
License: This is an open access article distributed under the terms of the Creative Commons Attribution License, which permits unrestricted use, distribution, reproduction and adaptation in any medium and for any purpose provided that it is properly attributed. For attribution, the original author(s), title, publication source (PeerJ) and either DOI or URL of the article must be cited.
License URL: https://creativecommons.org/licenses/by/4.0/

Keywords: Biomedical, LAMP, Hb Constant Spring, Hb Pakse, Thalassemia

Funding: HRH Princess Mahachakri Sirindhorn Medical Center, Faculty of Medicine, Srinakharinwirot University Contract No. 094/2567, 095/2567 Wittaya Jomoui and Rossarin Karnpean were supported by a research cluster in hematology and genetic diseases, as well as a research grant from HRH Princess Maha Chakri Sirindhorn Medical Center, Faculty of Medicine, Srinakharinwirot University Contract No. 443/2567 This study was supported by a research grant from HRH Princess Mahachakri Sirindhorn Medical Center, Faculty of Medicine, Srinakharinwirot University (Contract No. 094/2567, 095/2567). Wittaya Jomoui and Rossarin Karnpean were supported by a research cluster in hematology and genetic diseases, as well as a research grant from HRH Princess Maha Chakri Sirindhorn Medical Center, Faculty of Medicine, Srinakharinwirot University (Contract No. 443/2567). The funders had no role in study design, data collection and analysis, decision to publish, or preparation of the manuscript.

==============================
Thalassemia is a genetic disorder with significant prevalence in Southeast Asia, particularly in Thailand, where hemoglobin (Hb) Constant Spring (Hb CS) and hemoglobin Pakse (Hb PS) mutations are common. These mutations, resulting from stop codon alterations in the α2-globin gene, can lead to severe phenotypes such as non-deletional Hb H disease. This study aimed to develop and evaluate a novel colorimetric loop-mediated isothermal amplification (LAMP) assay for detecting Hb CS and Hb PS mutations. A total of 282 samples with several genotypes were recruited in the study. We developed LAMP assay, using a phenol red pH indicator, which provided visual detection of DNA amplification within 35 minutes at 65 °C. Both assays demonstrated a lower limit of detection of 0.625 ng/reaction and achieved 100% sensitivity and specificity across 282 DNA samples, validated against standard allele-specific polymerase chain reaction (PCR). Additionally, the assay’s minimal equipment requirements and cost-effectiveness make it suitable for use in community hospitals and large-scale screenings. The LAMP assay offers a rapid, accurate, and affordable alternative for Hb CS and Hb PS detection, addressing the challenges of managing thalassemia in genetically diverse and resource-limited regions like Thailand.

Introduction

Thalassemia is a genetic disorder characterized by a reduction or complete absence of globin chains, which are associated with mutations in the α-globin gene (chromosome 16) and/or the β-globin gene (chromosome 11) (Fucharoen et al., 1998; Weatherall & Clegg, 2001). Hemoglobin (Hb) Constant Spring (Hb CS, HBA2:c.427T > C) and Hb Pakse (HBA2:c.429A > T) are two of the non-deletional α-thalassemia 2 or α+-thalassemia (Panyasai, Satthakarn & Phasit, 2023). These mutations are arising from mutations in the termination codon of the α2-globin gene (TAA > CAA for Hb CS and TAA > TAT for Hb PS,) resulting in conversion of the stop codon to Gln and Tyr, respectively. The prevalence of Hb CS ranges from 1–6% in Southeast Asia and South China (Fucharoen & Winichagoon, 1987), while in northern and northeastern Thailand, it is found to be between 10–20% (Tritipsombut et al., 2012; Jomoui et al., 2015). The frequency of Hb PS ranges from 0.5 to 3.0% in this region (Panyasai, Satthakarn & Phasit, 2023; Pichanun et al., 2010). These two variants, characterized by elongated globin chains, have been removed and result in a small amount of 1–5% of total Hb. The heterozygous state of Hb CS and Hb PS is typically observed without clinical significance, whereas homozygosity for these mutations may be associated with a thalassemia intermedia phenotype, characterized by mild anemia, jaundice, and hepatosplenomegaly (Jomoui et al., 2015). However, hydrops fetalis syndrome caused by homozygous Hb CS has been reported in Thai and Chinese patients (Charoenkwan et al., 2006; He et al., 2016). Furthermore, interaction of Hb CS or Hb PS with α0-thalassemia can lead to severe Hb H disease commonly encountered in China and Southeast Asia (Jomoui et al., 2015). This Hb H disease is called non-deletional Hb H disease (Hb H-CS and Hb H-PS), which is usually more severe than deletional Hb H disease caused by deleted three α-globin genes (Panyasai, Satthakarn & Phasit, 2023).

Currently, laboratory investigations for detecting Hb CS and Hb PS are screened using hemoglobin analysis. However, the limitation of this method is its inability to differentiate between these two variants in Hb peaks due to their co-migration during electrophoresis. Furthermore, in some cases, Hb analysis may not detect these variant peaks. Therefore, allele-specific PCR or Sanger sequencing has been designed to confirm these two variants as a gold standard (Charoenwijitkul et al., 2019). The identification of Hb CS and Hb PS is crucial for the investigation of these variants in regions where thalassemia and hemoglobinopathies are prevalent, such as Thailand and Southeast Asia, in order to prevent severe Hb H disease in at-risk couples. However, this method has limitations, including time consumption, high costs, and dependence on specialists and specific equipment. As a result, it cannot be performed in general hospitals in Thailand, especially in community hospitals where resources are limited.

As an alternative molecular screening method, LAMP assays are employed for genetic screening due to their ability to amplify the LAMP target up to 109 times in approximately less than one hour, utilizing isothermal incubation. This results in high sensitivity and specificity, as demonstrated in several studies (Jomoui et al., 2022; Chomean et al., 2018; Wang et al., 2020). Moreover, interpretation can be conducted using various techniques, such as naked-eye colorimetric assessment, turbidity measurement, lateral flow dipstick analysis, gel electrophoresis, and fluorescent detection (Zhang, Lowe & Gooding, 2014). The LAMP colorimetric assay, utilizing a phenol red (pH indicator) system, was designed to facilitate visual detection of amplification based on proton production. This process occurs during the extensive LAMP reaction, causing a color change in the solution from pink to yellow (Jomoui et al., 2022). The detection of thalassemia mutations using LAMP techniques has been reported for α0-thalassemia (SEA and THAI deletions), α+-thalassemia (3.7 kb and 4.2 kb deletions), β-thalassemia mutations (654M, 41/42M, −28M, 17M, and 27/28M), and β-thalassemia deletions (3.5 kb and 45 kb deletions) (Jomoui et al., 2022; Chomean et al., 2018; Wang et al., 2020; Tepakhan & Jomoui, 2021). However, Hb CS and Hb PS have not been reported or developed with the LAMP technique. Thus, in this study, we aimed to develop and evaluate a new technique for detecting these based on a colorimetric LAMP assay in the high prevalence and genetic heterogeneity of thalassemia in Thailand.

Materials and Methods

Subjects and routine analysis

Ethical approval for this study was obtained from the Institutional Review Board of Srinakharinwirot University, Thailand (SWUEC-663007). In this study, we use leftover specimen from participants who written informed with broad consent form with the first collecting specimen. Leftover EDTA blood that investigated Hb analysis (Hb typing) and mean cell volume (MCV) < 82 fL was recruited in this study. A total of 282 samples were collected at MahaChakri Sirindhorn Medical Center, Faculty of Medicine, Srinakharinwirot University, Thailand. Furthermore, laboratory data in routine analysis such as complete blood count (CBC) and, Hb typing was recorded in this study. The hematological parameters were measured using hematology automation (Sysmex XN3000, Sysmex, Kobe, Japan). Hb typing (Hb analysis) was performed on capillary electrophoresis (Capillarys 2; Sebia, Lisses, France). Genomic DNA was extracted from blood leukocytes based on the magnetic bead extraction method using the Zybio Nucleic Acid extraction kit (WB-B-20:20T) (Zybio Inc., China) according to the manufacturer’s instructions. DNA analysis was performed on all samples with routine α-thalassemia gene detection. Identification of α-thalassemia alleles, including α0-thalassemia (SEA, THAI deletions), α+-thalassemia (3.7 kb and 4.2 kb deletions), and Hb CS & Hb PS mutation genes, was investigated in our laboratory using polymerase chain reaction (PCR) methods as described elsewhere (Charoenwijitkul et al., 2019; Tepakhan & Jomoui, 2021).

Development of LAMP-colorimetric assay

A novel molecular detection method based on a colorimetric LAMP with a phenol red assay was developed to detect Hb CS and Hb PS for the first time. All LAMP primer sets and probes in this study were designed using Primer Explorer V5 software (https://primerexplorer.eiken.co.jp/lampv5e/index.html). The common primer set including the outer primers (F3 and B3) and the inner primers (FIP and BIP) was shown in Table 1. Furthermore, two loop probes (LP) used in this study were designed for specific Hb CS (LP-CS) and Hb PS (LP-PS) as also shown in Table 1. These primers set and probes were protected according to petty patent submission number 2203001286 (Thailand). The protocol and data were adapted and collected as previously described in Tepakhan & Jomoui (2021), Karnpean et al. (2023) and Jomoui et al. (2024). The developed LAMP colorimetric phenol red assay was performed with reaction mixture and temperature as detailed in Table 1. The LAMP mixture was incubated at an isothermal temperature of 65 °C for 35 min using a Simpliamp thermal cycler (Thermo Fisher Scientific, Massachusetts, USA). The positive result was observed as yellow color by the naked eye compared with the negative result which was pink color. The protocol was uploaded via Jomoui (2025).

Table 1 Primers & probes set of LAMP colorimetric assays for identification of Hb Constant Spring and Hb Pakse’ genes.

LAMP assays	Primers & probes	Primer’s sequence (5′ > 3′)	Reaction mixture and temperature	
Hb constant spring (HBA2:c.427T > C)	F3	TCCCTGGACAAGTTC	Reaction mixture: A total of 12.5 µL contained 1.5 µL of (20–50) ng/µL genomic DNA, 6.25 µL of WarmStart® Colorimetric LAMP, 1.0 µL of 5M Betaine, 0.2 µL of 10 µM each primer of F3 and B3, 0.4 µL of 40 µM each primer of FIP and BIP, 1.5 µL of 10 µM of LP-CS, and the remainder distilled water. Isothermal temperature: 65 °C, 35 min	
B3	GTTGGCACATTCCGGGATA	
FIP	CGGGCAGGAGGAACGG CTACGAGCACCGTGCTGACCT	
BIP	CCTTGCACCGGCCCTT CCTGAGAACCCAGGCACACAC	
LP-CS	CAGCTTGACGGT	
Hb Pakse’ (HBA2:c.429A > T)	F3	TCCCTGGACAAGTTC	Reaction mixture: A total of 12.5 µL contained 1.5 µL of (20–50) ng/µL genomic DNA, 6.25 µL of WarmStart® Colorimetric LAMP, 1.0 µL of 5M Betaine, 0.2 µL of 10 µM each primer of F3 and B3, 0.4 µL of 40 µM each primer of FIP and BIP, 1.5 µL of 10 µM of LP-PS, and the remainder distilled water. Isothermal temperature: 65 °C, 35 min	
B3	GTTGGCACATTCCGGGATA	
FIP	CGGGCAGGAGGAACGG CTACGAGCACCGTGCTGACCT	
BIP	CCTTGCACCGGCCCTT CCTGAGAACCCAGGCACACAC	
LP-PS	CCAGCATAACGGTA	

The performance validation of the developed method was performed with sensitivity and specificity. To determine the sensitivity assay, the lower limit of detection (LOD) of the colorimetric LAMP assay for the detection of Hb CS and Hb PS was carried out using 4-fold serial dilution (DNA template range 40–0.156 ng/reaction), as shown in Fig. 1. The cross-reaction test or specificity of the developed LAMP assay for Hb CS was evaluated with several genotypes including, (αα/αα), (−α3.7/αα), (−α4.2/αα), (αCSα/αα), (αPSα/αα), (−α3.7/−α3.7), (αCSα/αCSα), (−α3.7/αCSα) (–SEA/αα), (–THAI/αα), (–SEA/−α3.7), (−SEA/αCSα), and no template control (NTC) (Fig. 2A). Whereas specificity of LAMP for Hb PS was evaluated with (αα/αα), (−α3.7/αα), (−α4.2/αα), (αCSα/αα), (αPSα/αα), (−α3.7/−α3.7), (αCSα/αCSα), (−α3.7/αPSα) (–SEA/αα), (–THAI/αα), (−SEA/−α3.7), and NTC (Fig. 2B).

Figure 1 Determination of the lower limit of detection (LOD) of the developed colorimetric LAMP assays for detecting Hb CS and Hb PS.

The DNA four-fold serial dilutions were started with 40 ng/reaction to 0.156 ng/reaction, including (A) Hb Constant Spring, (B) Hb Pakse assays.

Figure 2 The specificity assay of the colorimetric LAMP assays was demonstrated on samples with several thalassemia genotypes for detecting (A) Hb Constant Spring, (B) Hb Pakse assays.

A total of 282 DNA samples that were recruited with several thalassemia genotypes were examined in blinded trials with the colorimetric LAMP assay as described. The results are summarized in Table 2. The accuracy of the developed LAMP assays was compared with thalassemia genotypes based on conventional ASPCR or gap-PCR as the gold standard. Furthermore, the sensitivity, specificity, NPV, and PPV of the developed colorimetric LAMP assay to detect Hb CS and Hb PS in real clinical samples were calculated as shown in Tables 3 and 4.

Table 2 Thalassemia genotypes of 282 subjects with positive (P) and negative (N) in the LAMP colorimetric assays for Hb Constant Spring and Hb Pakse’.

P & N are positive and negative, respectively. ND indicates Hb CS or Hb PS has not been detected in Hb typing.

Genotypes	N(282)	Hb Types	Hb Fractions (%)	Hematological data	Colorimetric LAMP	
α-globin	β-globin			Hb A2/E	Hb F	Hb CS/PS	Hb (g/dL)	Hct (%)	MCV (fL)	MCH (pg)	Hb CS	Hb PS	
−SEA/−α3.7	βA/βA	2	A2A	0.9, 1.7	–	–	8.8, 9.6	30.3, 31.5	59.1, 51.7	17.2, 15.8	N	N	
βA/βA	1	A2AH	1.0	–	–	8.5	28.2	59.5	18	N	N	
βA/βA	1	A2ABart’H	1.4	1.2	–	9	26.8	56.7	19	N	N	
−SEA/αCSα	βA/βA	1	CSA2A	0.6	–	4.0	9.3	35.3	78.6	20.7	P (1)	N	
−SEA/αα	βA/βA	11	A2A	2.4 ± 0.1	–	–	10.5 ± 2.8	33.6 ± 8.5	64.9 ± 4.9	20.3 ± 1.9	N	N	
βT/βA	1	A2A	5.0	0.9	–	13.9	44.4	68.8	21.6	N	N	
βE/βA	5	EA	18.6 ± 2.6	–	–	11.3 ± 4.9	39.6 ± 7.9	62.6 ± 5.2	19.8 ± 2.0	N	N	
βE/βE	2	EE	100.0, 100.0	–	–	11.2, 9.5	33.0, 19.4	60.4, 48	20.5, 15.5	N	N	
−α3.7/−α3.7	βA/βA	7	A2A	2.5 ± 0.2	–	–	12.5 ± 2.4	36.8 ± 5.8	69.0 ± 5.1	21.9 ± 1.4	N	N	
βE/βA	6	EA	19.9 ± 0.9	–	–	11.7 ± 1.0	36.5 ± 3.4	69.7 ± 3.4	22.5 ± 1.4	N	N	
βE/βE	1	EE	100	–	–	8.2	25.1	53.2	17.4	N	N	
−α3.7/−α4.2	βA/βA	1	A2A	2.5	–	–	11.9	37.4	72.6	23.1	N	N	
−α3.7/αCSα	βA/βA	3	CSA2A	1.8 ± 0.2	–	0.9 ± 0.2	12.7 ± 1.8	41.2 ± 5.4	70.1 ± 2.4	21.6 ± 1.1	P (3)	N	
βE/βE	1	EE	98.1	1.9	ND	9.7	28.5	66.9	22.8	P (1)	N	
αCSα/αCSα	βA/βA	1	CSA2A	1.4	2.8	4.4	9.3	30.1	74.5	23	P (1)	N	
−α3.7/αα	βA/βA	49	A2A	2.6 ± 0.2	–	–	12.8 ± 2.3	39.7 ± 6.4	78.3 ± 3.7	25.2 ± 2.1	N	N	
βT/βA	3	A2A	5.6 ± 0.5	–	–	10.7 ± 0.4	33.8 ± 0.8	67.7 ± 2.5	21.5 ± 1.0	N	N	
βE/βA	27	EA	26.1 ± 1.5	0.5 ± 0.3	–	12.7 ± 1.9	38.4 ± 5.4	77.2 ± 6.2	25.7 ± 2.4	N	N	
βE/βE	1	EE	98.2	1.8	–	10.1	29.3	61.3	21.1	N	N	
−α4.2/αα	βA/βA	3	A2A	2.5 ± 0.3	–	–	13.6 ± 0.3	41.4 ± 0.3	80.9 ± 3.6	26.7 ± 1.1	N	N	
αCSα/αα	βA/βA	10	CSA2A	2.1 ± 0.4	0.6 ± 0.1	0.5 ± 0.2	12.4 ± 2.2	38.1 ± 6.9	76.1 ± 3.2	24.2 ± 1.2	P (10)	N	
βA/βA	1	A2A	2.1	–	ND	11.5	35.5	81.5	26.6	P (1)	N	
βT/βA	1	A2A	5.3	0.5	ND	12.0	37.1	76.5	24.7	P (1)	N	
βE/βA	3	EA	26.3 ± 1.1	–	ND	12.9 ± 1.6	40.8 ± 7.4	80.8 ± 4.8	26.9 ± 2.0	P (3)	N	
βE/βA	2	CSEA	15.3, 26.3	–	0.5, 0.3	10.6, 14.6	34.8, 45.7	76.0, 78.3	23.8, 25.0	P (2)	N	
βE/βE	1	EE	98.9	1.1	ND	9.4	28.8	61.8	20.2	P (1)	N	
αPSα/αα	βA/βA	2	A2A	1.8, 2.2	–	ND	5.8, 12.7	20.0, 39.0	60.8, 79.1	17.6, 25.8	N	P (2)	
βE/βA	3	EA	25.5 ± 0.8	–	ND	13.8 ± 2.2	42.1 ± 7.0	76.9 ± 0.9	25.2 ± 0.5	N	P (3)	
αα/αα	βA/βA	41	A2A	2.6 ± 0.4	1.3 ± 0.3	–	12.2 ± 2.3	37.2 ± 6.4	78.7 ± 5.2	26.0 ± 2.7	N	N	
βT/βA	7	A2A	5.1 ± 0.7	2.4 ± 2.0	–	11.5 ± 3.1	35.7 ± 8.7	66.5 ± 9.5	21.5 ± 3.8	N	N	
βE/βA	77	EA	28.2 ± 1.5	–	–	11.8 ± 2.3	36.1 ± 6.8	73.7 ± 6.4	24.1 ± 2.7	N	N	
βE/βE	6	EE	97.0 ± 2.3	4.9 ± 3.3	–	10.8 ± 1.3	32.1 ± 4.4	58.0 ± 3.8	19.5 ± 1.9	N	N	
βE/β+	1	EFA	48.4	4.1	–	10.5	33.2	51.6	16.3	N	N	

Table 3 The sensitivity, specificity, positive and negative predictive values of colorimetric LAMP assays compared with allele specific-PCR analysis as a gold standard for detecting Hb Constant Spring.

Colorimetric LAMP	Allele specific-PCR analysis	Total	
	Positive	Negative		
Positive	24	0	24	
Negative	0	258	258	
Total	24	258	282	
Notes.

Sensitivity = (24/24) × 100 = 100%.

Specificity = (258/258) × 100 = 100%.

Positive predictive value = (24/24) × 100 = 100%.

Negative predictive value = (258/258) × 100 = 100%.

Table 4 The sensitivity, specificity, positive and negative predictive values of colorimetric LAMP assays compared with allele specific-PCR analysis as a gold standard for detecting Hb Pakse’.

Colorimetric LAMP	Allele specific-PCR analysis	Total	
	Positive	Negative		
Positive	5	0	5	
Negative	0	277	277	
Total	5	277	282	
Notes.

Sensitivity = (5/5) × 100 = 100%.

Specificity = (277/277) × 100 = 100%.

Positive predictive value = (5/5) × 100 = 100%.

Negative predictive value = (277/277) × 100 = 100%.

Results

Table 2 represents a total of 282 samples with several thalassemia genotypes in this study. Twelve genotypes based on α-globin gene defect were illustrated co-inheritance with several β-thalassemia and Hb E. These data confirmed the high heterogeneity and prevalence of thalassemia and hemoglobinopathies in this region. Heterozygous α+-thalassemia (−α3.7/αα) was found highest in this study. In this study, Hb CS gene was recruited with 24 samples contained (–SEA/αCSα, n = 1), (−α3.7/αCSα, n = 4), (αCSα/αCSα, n = 1), (αCSα/αα, n = 18) whereas Hb PS included with (αPSα/αα, n = 5). The hematological data was presented with Hb (g/dL), Hct (%), MCV (fL), and MCH (pg) in mean with SD. This finding was correlated with severity based on genotype and phenotype. Mostly anemia condition was observed in Hb H disease, while minimal anemia could be found in heterozygous or homozygous states of α+-thalassemia. Furthermore, low MCV and MCH would be correlated with the degree of thalassemia genotypes and phenotypes.

At Hb analysis, thalassemia phenotypes were categorized using a capillary electrophoresis (CE) with Hb types such as Hb H disease, Hb variant, and β-thalassemia trait or disease. However, unstable globin chains or unstable Hb globin in some cases could not be detected by this machine. Among five cases of Hb H disease, three samples had an absent Hb H peak in the results. However, these groups mostly were observed with anemia and low Hb A2 levels (<2%). Of the 24 samples carrying Hb CS gene, seven samples (29.17%) had no Hb CS/PS peak in the electropherogram. Moreover, all five samples with the Hb PS gene (100%) had no found Hb CS/PS peak in the electropherogram. We calculated sensitivity and specificity for detection of Hb CS/PS using CE in this study was 58.62% and 100.00%, respectively.

The analytical performance validation of the developed colorimetric LAMP assays was illustrated in Figs. 1 and 2. To assess the sensitivity of the developed LAMP assays, LOD values were determined, as shown in Fig. 1. The LOD of colorimetric LAMP assays was 0.625 ng/reaction in both Hb CS (Fig. 1A) and Hb PS genes (Fig. 1B). Whereas, the analytical specificity of the developed LAMP assays had no cross-reactivity with other thalassemia genotypes.

The prospective clinical sample validation was evaluated with 282 blinded samples with several genotypes. The results of colorimetric LAMP assays to detect Hb CS and Hb PS genes were summarized in Table 2. Of the 282 samples, all 24 samples with the Hb CS gene were found positive in the colorimetric LAMP assay for Hb CS genes. The remaining 258 samples with no Hb CS gene were negative in the same assay. This 100% concordance result between the colorimetric LAMP assay and AS-PCR as the gold standard method for the detection of the Hb CS gene means the sensitivity, specificity, NPV, and PPV in clinical diagnosis were calculated as 100% in all parameters, as shown in Table 3. The developed colorimetric LAMP assay for Hb PS genes was also validated with blinded 282 samples. All five samples carrying Hb PS gene were found positive in the developed LAMP assays for Hb PS. The remaining 277 with no Hb PS gene had no represented cross-reaction in the assays. The calculation of the sensitivity, specificity, NPV, and PPV was also 100% in all parameters, respectively (Table 4). We could demonstrate our two developed LAMP assays with no false positives or false negatives in any of them.

We also assessed the test effectiveness, which compared the developed LAMPs assay and AS-PCR for detecting Hb CS and Hb PS. The cost per test of the developed LAMPs assay and AS-PCR based on reagent is currently USD 4.0 and 3.0, respectively. Time consumed for the developed LAMP assay was 35 min, whereas AS-PCR technique was 150 min. The developed LAMP required only dry bath or thermocycler, while AS-PCR required a thermocycler, gel electrophoresis, UV camera gel documentation.

Discussion

Hb CS and Hb PS are prevalent in populations across Thailand, Laos, Cambodia, and southern China, with rates of Hb CS reaching 10–20% in northern Thailand and Hb PS observed at 0.5–3.0% in the region (Fucharoen & Winichagoon, 1987; Tritipsombut et al., 2012; Jomoui et al., 2015; Pichanun et al., 2010). The clinical relevance of these mutations is underscored by their association with non-deletional Hb H disease when co-inherited with α0-thalassemia. This form of Hb H disease often presents with more severe anemia, jaundice, and hepatosplenomegaly compared to deletional Hb H disease. However, non-deletional Hb H disease in Thailand was predominantly only Hb H-CS disease whereas the rare type Hb H-PS was also reported in this region (Jomoui et al., 2020). Furthermore, homozygosity for Hb CS has been reported to result in hydrops fetalis, highlighting the critical need for carrier screening to prevent severe outcomes (Charoenkwan et al., 2006; He et al., 2016).

In Thailand, thalassemia prevention programs have significantly reduced the incidence of severe cases by implementing carrier screening and prenatal diagnostic services. CE and other hemoglobin analysis techniques, such as high-performance liquid chromatography (HPLC), are widely used for thalassemia screening due to their ability to separate and identify hemoglobin variants based on their migration patterns (Panyasai, Satthakarn & Phasit, 2023; Srivorakun et al., 2014). However, these methods have limitations in detecting unstable or low-concentration hemoglobins like Hb CS and Hb PS. In this study, CE failed to detect Hb CS in 29.17% of cases and Hb PS in 100% of cases, reflecting the inherent weaknesses of electrophoretic techniques in identifying mutations with minimal hemoglobin production. However, Panyasai, Satthakarn & Phasit (2023) have also reported that CE failed to detect Hb CS in 6.6%. Unstable hemoglobin variants are prone to structural alterations that can compromise their integrity. If blood samples containing these variants are left for extended periods before testing, factors such as temperature fluctuations, changes in pH, or enzymatic activity may cause the hemoglobin molecules to degrade. This degradation can lead to the loss of Hb peak or the appearance of abnormal Hb in the electrophoresis.

As shown in Figs. 1 and 2, we demonstrated that the colorimetric LAMP assays developed to detect Hb CS and Hb PS mutations are straightforward to use and exhibit high sensitivity and specificity. The lower limit of detection (LOD) for these assays was determined to be 0.625 ng/reaction, which ensures their applicability for general DNA sample types. The achieved LOD of 0.625 ng/reaction is consistent with or superior to other molecular diagnostic methods, such as allele-specific PCR (AS-PCR) and qPCR, which often require higher DNA input for reliable results. Typically, DNA extracted from EDTA blood is present at concentrations exceeding 10 ng/µL, making it compatible with the developed assays (Tepakhan & Jomoui, 2021; Psifidi et al., 2015). In this study, DNA concentrations ranged from 20 to 50 ng/µL, further confirming the feasibility of using these assays in clinical settings. To ensure assay specificity, we validated the LAMP primers and assay conditions against other thalassemia genes and wild-type alleles, with no cross-reactivity observed, as shown in Fig. 2. The specificity of the developed LAMP is particularly important given the genetic complexity of thalassemia in Southeast Asia. Furthermore, the loop probes used in this study were specifically designed for the Hb CS and Hb PS mutations, allowing for the amplification of only the targeted sequences. This approach reduces the detection time to just 35 min, which is better than other reported LAMP-based methods for thalassemia detection (Jomoui et al., 2022; Chomean et al., 2018; Wang et al., 2020; Jomoui et al., 2024; Karnpean et al., 2023).

Previous studies have documented the use of colorimetric LAMP assays for detecting various thalassemia genes, including α0-thalassemia (SEA deletions). However, one concern is the occurrence of false positives in some cases (Jomoui et al., 2022). Furthermore, detection of thalassemia mutations using LAMP techniques has previously been reported for various α-thalassemia deletions, including SEA and THAI (α0-thalassemia) and 3.7 kb and 4.2 kb deletions (α+-thalassemia), as well as several β-thalassemia point mutations (Jomoui et al., 2022; Chomean et al., 2018; Wang et al., 2020; Tepakhan & Jomoui, 2021). Although LAMP technology has been successfully applied to detect common thalassemia, its application to non-deletional α-thalassemia mutations such as Hb CS and Hb PS has not previously been reported. This may be due to the technical challenge of designing primers that can distinguish single nucleotide changes where these mutations occur. Unlike large deletions, which offer distinct target regions for primer binding, point mutations like Hb CS and Hb PS require highly specific primer design and optimization to ensure accuracy and avoid cross-reactivity. LAMP assay is known for its high sensitivity; this sensitivity can sometimes lead to the amplification of non-target sequences, which may result in false positives. The high sensitivity of the LAMP method means that even low levels of DNA, or closely related genetic sequences, can be detected, potentially amplifying unintended products. To mitigate this, optimization of reaction conditions and the use of specific primers are essential to minimize non-specific amplification. Despite these challenges, the simplicity and accessibility of the LAMP assay make it a promising tool for thalassemia detection, provided that careful validation is conducted to ensure its accuracy and reliability (Jomoui et al., 2022; Wang et al., 2020).

To assess the clinical utility of the developed colorimetric LAMP assays, we evaluated 282 DNA samples representing various genotypes prevalent in this region. As shown in Table 2, the results of the colorimetric LAMP assays demonstrated 100% concordance with the conventional AS-PCR method for detecting Hb CS and Hb PS mutations. The calculated sensitivity, specificity, negative predictive value (NPV), and positive predictive value (PPV) were all 100% for both assays, as summarized in Tables 3 and 4. These findings underscore the robustness of the developed colorimetric LAMP assays, confirming their diagnostic accuracy in detecting Hb CS and Hb PS mutations. This is particularly significant in regions like Southeast Asia, where genetic heterogeneity and the high prevalence of thalassemia mutations demand precise and efficient diagnostic tools (Fucharoen et al., 1998; Weatherall & Clegg, 2001; Jomoui et al., 2015). The high sensitivity and specificity ensure that the assays are highly effective in minimizing both false-negative and false-positive results. The developed assays have the potential to significantly enhance early screening efforts for Hb CS and Hb PS, reducing the incidence of severe outcomes such as Hb H disease (Jomoui et al., 2020).

Although this study is limited by the relatively small number of positive samples for Hb CS (n = 24) and Hb PS (n = 5), which may restrict the generalizability of the LAMP assay’s performance metrics, 282 samples with various genotypes were evaluated in a blinded manner to confirm both specificity and sensitivity. Furthermore, validation in larger, population-based studies is necessary to establish LAMP as a reliable alternative for routine screening. Furthermore, another limitation of the current LAMP assay is its inability to distinguish between heterozygous and homozygous genotypes for these mutations. This distinction is clinically significant, particularly in prenatal screening and genetic counseling, where zygosity influences disease severity and management decisions.

Thailand and Southeast Asia challenges in managing thalassemia due to the region’s high genetic heterogeneity and varying healthcare capacities. The developed LAMP assays address these challenges by offering several advantages. Its accessibility is a key benefit, as it requires minimal equipment, making it suitable for use in community hospitals and rural healthcare settings. Additionally, the simplicity of the assay, with its visual detection system, allows for easy interpretation of results, reducing the dependency on highly trained personnel. The cost-effectiveness of the developed LAMP assays is another significant advantage, priced at just USD 4.0 per test, making it affordable, particularly when used for large-scale population screenings. Furthermore, the assay’s efficiency is remarkable, with a runtime of only 35 min, enabling rapid decision-making, which is essential for prenatal and carrier screening programs. These features make the LAMP assay an invaluable tool for managing thalassemia in regions with limited resources.

Conclusion

In this study, a novel colorimetric LAMP assay was developed and validated for the rapid detection of hemoglobin Constant Spring (Hb CS) and hemoglobin Pakse’ (Hb PS) mutations, achieving 100% sensitivity and specificity compared to conventional allele-specific PCR. With its cost-effectiveness, minimal equipment requirements, and rapid results within 35 min, the LAMP assay is particularly suited for use in resource-limited settings, including community hospitals and rural healthcare facilities.

Supplemental Information

Supplemental Information 1 Raw data (Hematological and molecular testing)

Additional Information and Declarations

Competing Interests

Author Contributions

Human Ethics

Data Availability

The authors declare there are no competing interests.

Kasama Wongprachum conceived and designed the experiments, analyzed the data, prepared figures and/or tables, authored or reviewed drafts of the article, and approved the final draft.

Nichakan Thitipoomdecha performed the experiments, analyzed the data, prepared figures and/or tables, and approved the final draft.

Phakkamon Ananratanakit performed the experiments, analyzed the data, prepared figures and/or tables, and approved the final draft.

Wattanakit Prakobkul performed the experiments, analyzed the data, prepared figures and/or tables, and approved the final draft.

Unchasa Angkuranak performed the experiments, analyzed the data, prepared figures and/or tables, and approved the final draft.

Nitchagan Sawangkul performed the experiments, analyzed the data, prepared figures and/or tables, and approved the final draft.

Prapaporn Panichchob performed the experiments, analyzed the data, prepared figures and/or tables, authored or reviewed drafts of the article, and approved the final draft.

Rossarin Karnpean performed the experiments, analyzed the data, prepared figures and/or tables, authored or reviewed drafts of the article, and approved the final draft.

Wittaya Jomoui conceived and designed the experiments, performed the experiments, analyzed the data, prepared figures and/or tables, authored or reviewed drafts of the article, and approved the final draft.

The following information was supplied relating to ethical approvals (i.e., approving body and any reference numbers):

Ethical approval for this study was obtained from the Institutional Review Board of Srinakharinwirot University, Thailand (SWUEC-663007).

The following information was supplied regarding data availability:

The raw data of hematological and molecular testing are available in the Supplemental File.

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
