# Peer review of "Utilizing loop-mediated isothermal amplification (LAMP) for detecting hemoglobin Constant Spring and hemoglobin Pakse mutations amidst the high prevalence and genetic heterogeneity of thalassemia in Thailand"

_PeerJ, doi:10.7717/peerj.19687_

## Round 0.1 · original submission · Minor Revisions

Could you please address the concerns raised by the reviewers.

**Language Note:** The review process has identified that the English language must be improved. PeerJ can provide language editing services - please contact us at [email protected] for pricing (be sure to provide your manuscript number and title). Alternatively, you should make your own arrangements to improve the language quality and provide details in your response letter. – PeerJ Staff

Reviewer 1 ·

Basic reporting

The language is clear, with a well-defined aim and relevance of the study.

Experimental design

The methodology is well described.

Validity of the findings

The sample number (n=284) is not adequate to establish LAMP PCR to be a reliable alternate method of detection. 24 samples are true positive for HbCS, and 5 samples are true positive for Hb Pakse. Whether these numbers are adequate to demonstrate a reliable sensitivity could be a major concern.

Additional comments

Even though the sample size is limited, considering the potential application of the study, I would give a positive recommendation.

Reviewer 2 ·

Basic reporting

In this study, the authors have used the LAMP technique to detect the hemoglobin Constant Spring and hemoglobin Pakse9 mutations.
1) It is better that the manuscript be revised by a native English speaker.
2) Literature references are sufficient except for the discussion.
3) Figures and tables are good.

Experimental design

Experiment design and methods have been well expressed.

Validity of the findings

Results are explained good, but I believe that it is better to compare their results with similar studies more in the discussion section.

Reviewer 3 ·

Basic reporting

This manuscript describes the novel colorimetric LAMP technique used for the detection of Hb Constant Spring and Hb Pakse. The manuscript was well prepared, and the study was well designed. The only suggestion for this manuscript is to check for English grammar and tense, as some errors in tense were found in the content.

Experimental design

The study was well experimentally designed.

Validity of the findings

The findings were valid and reliable.

Additional comments

Only English grammar proofreading was required.

Reviewer 4 ·

Basic reporting

Well-structured background on genetic diversity and clinical relevance is provided. Most of the references cited in the text are up to date. Figures and tables are clear for review. However, some sentence constructions need revision for clarity. The authors are also suggested to add information on why LAMP has not previously been applied to these specific mutations, despite its use for beta- and deletional alpha-thalassemia, to highlight the knowledge gap.

Experimental design

The aim of the study is clearly defined. However, there are some concerns that need to be addressed by the authors:

i) Please justify why the sample with different genotypes tested for the LAMP assay in Figure 2 is not equal. Is there any specific reason why some genotypes are not included in the LAMP assay for Hb CS and vice versa?

ii) The use of phenol red as a pH indicator in the LAMP assay is valid, but its selection over other detection methods is not justified.

iii) Can the authors explain how the specificity of self-designed primers was ensured in this study? For example, how do the loop primers achieve specificity for Hb CS (TAA > CAA) and Hb PS (TAA > TAT), given the high sequence similarity?

iv) The authors also did not mention clearly how many technical replicates are included for the LOD of the calorimetric LAMP assay.

Validity of the findings

i) Did the authors validate the amplification of the LAMP assay in Figure 2 using gel electrophoresis? If yes, it would be great if the authors could provide the gel pictures for validation.

ii) The study emphasized that the developed LAMP assay was able to detect DNA down to 0.625 ng/reaction. However, did the authors test the assay with degraded DNA samples, which are often encountered in field conditions?

iii) Is the assay able to distinguish heterozygous from homozygous, which is clinically relevant in prenatal screening? If not, it would be great if the authors could address it as one of the limitations of this LAMP assay.

Reviewer 5 ·

Basic reporting

The manuscript is written in short and clear language. The references used are appropriate to the topic. The results of the study bring important meaning to the diagnosis of hemoglobinopathies

Experimental design

In general, the research design used is appropriate.
There are some things that need to be added:
- How to rule out HbE in the study sample
- Method of DNA testing with PCR
- Reason for choosing LOD 0.625 ng
The details can be seen in the attached manuscript review.

Validity of the findings

The research data is clearly presented. The results are novel and important in helping to diagnose hemoglobinopathies.

Annotated reviews are not available for download in order to protect the identity of reviewers who chose to remain anonymous.

---

## Round 0.2 · accepted · Accept

The authors have addressed all concerns raised by reviewers.